Continuous bubble streams for controlling marine biofouling on static artificial structures

Hopkins Grant A. grant.hopkins@cawthron.org.nz 1
Gilbertson Fletcher 1
Floerl Oli 1
Casanovas Paula 1
Pine Matt 2
Cahill Patrick 1
1 Cawthron Institute , Nelson , New Zealand
2 Department of Biology, University of Victoria , B.C. Canada
Anderson Todd
Electronic publication date: 2021 Apr 30
Publication date: 2021
Volume: 9
Electronic Location ID: e11323
Received 2020 Nov 24; Accepted 2021 Mar 31
Copyright: ©2021 Hopkins et al.
Copyright year: 2021
Copyright holder: Hopkins et al.
License: This is an open access article distributed under the terms of the Creative Commons Attribution License, which permits unrestricted use, distribution, reproduction and adaptation in any medium and for any purpose provided that it is properly attributed. For attribution, the original author(s), title, publication source (PeerJ) and either DOI or URL of the article must be cited.
License URL: https://creativecommons.org/licenses/by/4.0/

Keywords: Air bubbles, Marinas, Static structures, Settlement, Treatment

Funding: Cawthron’s Internal Investment Fund and New Zealand’s Ministry of Business, Innovation and Employment funding CAWX1904 The publication of this work was supported by Cawthron’s Internal Investment Fund and New Zealand’s Ministry of Business, Innovation and Employment funding (CAWX1904—A toolbox to underpin and enable tomorrow’s marine biosecurity system). The funders had no role in study design, data collection and analysis, decision to publish, or preparation of the manuscript.

==============================
Biofouling accumulation is not proactively managed on most marine static artificial structures (SAS) due to the lack of effective options presently available. We describe a series of laboratory and field trials that examine the efficacy of continuous bubble streams in maintaining SAS free of macroscopic biofouling and demonstrate that this treatment approach is effective on surface types commonly used in the marine environment. At least two mechanisms were shown to be at play: the disruption of settlement created by the bubble stream, and the scouring of recently settled larvae through shear stress. Field trials conducted over a one-year period identified fouling on diffusers as a major issue to long-term treatment applications. Field measurements suggest that noise associated with surface mounted air blowers and sub-surface diffusers will be highly localised and of low environmental risk. Future studies should aim to develop and test systems at an operational scale.

Introduction

Harbours throughout the world have been heavily modified and contain a vast surface area of static artificial structures (SAS) associated with ports, marinas and other facilities (Dugan et al., 2011; Firth et al., 2016). These surfaces are quickly colonised by biological fouling (biofouling) communities that are often dominated by non-indigenous species (Glasby et al., 2007; Airoldi et al., 2015). For some structures, fouling can be costly to manage and lead to a range of undesirable outcomes, including biocorrosion (e.g., on oil rigs and wind farms), increased loading and hydrodynamic drag (e.g., for pontoons and warps), and crop or stock losses in aquaculture (Bannister et al., 2019). For structures like wharf piles and marina pontoons, the financial incentives to prevent or remove fouling are less evident and fouling is often left unmanaged.

The establishment and spread of marine pests within coastal systems is closely monitored in countries like New Zealand, Australia and the United States, and ports and marinas are typically the focal points for systematic surveys. However, SAS associated with ports and marinas are rarely actively maintained or treated to prevent the establishment and proliferation of biofouling, except in the case of pest eradication and population control efforts (Forrest & Hopkins, 2013). Biofouling accumulations on SAS are generally extensive and result in highly elevated recruitment rates to resident or visiting vessels or other submerged structures (Floerl & Inglis, 2003; Floerl & Inglis, 2005). Reducing biofouling on SAS will reduce the need for periodic and expensive biofouling-related maintenance of infrastructure. This will reduce the likelihood of marine pest establishment in vector hubs (i.e., act as a barrier to new incursions), and interrupt subsequent spread by reducing resident vessel fouling rates (i.e., through reduced propagule supply). These indirect benefits of managing biofouling on SAS associated with ports and marinas, as well as other artificial environments such as aquaculture farms, are in addition to direct improvements in structural integrity and longevity by minimising biofouling-related hydrodynamic drag and microbially-induced corrosion.

The arsenal of tools to manage SAS biofouling is limited, with mechanical approaches (e.g., water blasters, scrapers, mechanical brushes) the most widely available. The present suite of antifouling paints available on the market are generally not amenable to use on SAS, with the three main classes of soluble-matrix coatings, self-polishing copolymers, and silicone- or fluoropolymer-fouling release coatings all requiring water flows above threshold values to remain effective. In addition, periodic removal of SAS for antifouling coating renewal would in many cases be unfeasible. Therefore, the most common approaches to fouling management on SAS is to do nothing and accept the consequences or to rely on physical removal methods.

Several studies have highlighted the potential effectiveness of bubble streams (or curtains) in controlling biofouling accumulation on artificial surfaces in the marine environment. Scardino, Fletcher & Lewis (2009) tested the efficacy of bubble streams over acrylic and fouling release panels held in a V-shaped frame to replicate the submerged portion of a vessel hull, followed by a field trial on a hull section of a commercial vessel. Significant differences were observed between treated and untreated panels; however, some macrofouling accumulation (e.g., hydroids) occurred despite treatment. Similarly, the hull section of the vessel being treated was largely devoid of macrofouling (covering only 10% of the experimental surface area) compared to the adjacent non-treated areas of the hull (88% macrofouling cover), although a slime layer still developed. Bullard, Shumway & Davis (2010) investigated the efficacy of continuous bubble exposure applied to PVC and concrete panels deployed over relatively short periods (1- and 4-week deployments) at three sites with different seabed communities. Like Scardino, Fletcher & Lewis (2009), rates of macrofouling development were significantly lower on panels treated with bubbles (4% of that on controls after 4 weeks). Lowen et al. (2016) tested the effects of bubble streams and suspended particles on the settlement and survivorship of early life-stages of Ciona intestinalis. Larval settlement was effective at flows >10 L min−1 (with 0% recruitment at 20 L min−1), but juveniles that had settled for 21 days were resilient to treatment. The authors concluded that continuous treatment would be required to prevent Ciona establishment on structures.

In contrast to these earlier investigations, which involved exposing surfaces to a ‘cloud’ of bubbles, Menesses et al. (2017) undertook laboratory and field investigations into the minimum shear stress required for a continuous single bubble stream to remove biofouling. This study concluded that shear stress forces of around 0.01 Pa are required to prevent biofouling accumulation.

Despite the early promise shown by bubble stream approaches to biofouling management described above, there has been no real-world uptake by biofouling managers or technology developers. This lack of uptake is likely because key knowledge gaps remain. Firstly, the mode or mechanism of treatment is unclear. Studies to date have primarily focused on shear stresses created by bubble streams dislodging adhered fouling organisms (e.g., Menesses et al., 2017). We hypothesise that a continuous stream of bubbles also has the potential to disrupt the settlement process. Here we define disruption as either: (1) creating a physical barrier, by the movement of many bubbles or the formation of a large bubble on the experimental surfaces, or (2) the instantaneous removal of larvae from the surface before they are able to settle. Secondly, there is a lack of guidance around crucial operational requirements for effective treatment, such as bubble stream flow rates, frequency and duration of application. Artificial surfaces in the marine environment include a diversity of surface types (e.g., concrete, polyethylene, wood) and orientations, and it is unclear how bubble stream performance could be impacted by this diversity. Finally, although bubble streams likely present a lower environmental risk than traditional antifouling coatings (Bullard, Shumway & Davis, 2010), there is a growing body of evidence identifying detrimental impacts of underwater noise pollution on marine mammals and fish (see Jones, 2019; Slabbekoorn et al., 2010). Quantifying noise levels generated by bubble-based treatment systems is therefore necessary prior to large-scale implementation.

The present study describes a series of laboratory and field trials that collectively aimed to address the knowledge gaps identified above. We investigate whether treatment efficacy could be enhanced through changing the surface type of SAS and/or angle of application. We also investigate underwater noise emissions generated from our field test system, as this would arguably be the key environmental risk factor associated with widespread use. Collectively, our experiments explore whether bubble streams could be an effective, environmentally benign treatment method for SAS and stationary vessels with the intent to facilitate the uptake of bubble-based biofouling prevention technologies.

Materials and Methods

Laboratory trials

The purpose of the laboratory experiments described below was to explore mechanisms of action, as well as inform decisions around treatment parameters applied in subsequent field trials (e.g., flow rate, angle of application, surface type).

Testing system and surface types

Laboratory trials were undertaken in a temperature-controlled room (18 ± 1 °C). The experimental apparatus comprised a 478-L tank (L × W × H: 1115 × 715 × 600 mm) with two Hyotube™ Series 9 fine bubble diffusers (model number 9-200EP-KPH-4; pore size = 1 mm; Ecologix Technology) placed on the bottom of the tank (Fig. 1). The diffusers were powered by a blower (K04 MS MOR 1.1kW; FPZ Blower Technology). An aluminium frame, fixed ca. 400 mm directly above the diffusers, held the experimental panels in place (horizontally) during treatment. Flow rates from the diffusers were manipulated by venting excess air from the blower to the atmosphere via a ball valve. Air flow rates were estimated by inverting a 1000-mL measuring cylinder filled with water above the diffusers at the depth of the experimental panels, and the time taken to displace 500 mL of water recorded. Prior to each trial, flows were measured and adjusted to achieve either high (2.6 L h−1 cm−2), medium (1.7 L h−1 cm−2), or low (0.9 L h−1 cm−2) flow rates. Estimated shear stresses created by the bubble streams within the testing system under these different flow scenarios are provided in the Supplementary Material (S1) . Our system also included a second control tank that was identical to the treatment tank except it did not have air diffusers or air pumped into the water. Two panel types were used in the experiments: plain black acrylic panels with a matt texture finish (ACR), and acrylic panels professionally coated with the fouling release coating Intersleek 1100SR (FR-IS1100, International Paint). Panel dimensions were 200 × 150 mm (W × L).

Figure 1 Schematic of the treatment system used in both the scouring (A) and settlement disruption (B) trials.

The treatment tank contained two diffusers (side-by-side, 200 mm apart) that were powered by a 1.1 kw blower (C). Aluminium racks were used to hold experimental panels approximately 400 mm above the diffusers. The control tank used in the disruption experiments was configured the same as the treatment tank, minus the diffusers and associated pipework. Experimental details for the scouring and disruption trials are shown. Note that Ciona larvae could not be reliably settled onto the fouling release panels (Intersleek 1100SR, FR-IS1100) so were excluded from the analyses. ACR = acrylic panel (200 × 150 mm).

Model organisms

Two model organisms were used to explore bubble stream modes of action: the Pacific transparent sea squirt Ciona savignyi and the Pacific oyster Crassostrea gigas. Both species are non-indigenous to New Zealand and are considered nuisance fouling organisms. In our experiments, Ciona larvae were ready to settle onto experimental surfaces within 24 h of hatching, whereas Crassostrea larvae were reared for 17 d prior to settling. Detailed spawning, rearing and larvae settling procedures are provided in the Supplementary Material (S2).

Mechanisms of action: scouring

The ability of bubble streams to ‘scour’ settled larvae was investigated by pre-settling Ciona (an estimated density of 0.5 individuals cm−2 based on counts from control panels) and Crassostrea larvae (0.3 individuals cm−2) onto experimental panels and quantifying removal efficacy for variations of surface type (fixed, 2 levels: ACR and FR-IS1100), settlement time prior to treatment (2 levels, fixed: 3 and 120 h), and flow rates (4 levels, fixed: 0, 0.9, 1.7 and 2.5 L h−1 cm−2). These factors were assessed in a fully-crossed experimental design with n = 5 experimental panels for each treatment combination (Fig. 1). Larvae settlement times, flow rates and the treatment duration parameters used in these experiments were selected based on extensive pilot work (not presented in this paper) using the same treatment system and model species.

The ACR and FR-IS1100 panels were treated simultaneously across the four flow rates (order randomised) and two settlement times (3 and 120 h), with bubble stream treatment lasting 10 min. Panels settled with Ciona larvae for 3 h were placed in a 300-L recirculating holding tank for 5 days following treatment (i.e., until the 120 h settlement panels had also been treated). Ciona larvae are lecithotrophic and therefore did not requiring feeding within this timeframe. By contrast, the 3 h Crassostrea panels were placed in a 300 L non-recirculating aerated tank following treatment, and were fed approximately 5 L of Isochrysis galbana (approx. 10 million cells mL−1) daily for 5 days, with a 50% water change undertaken after 2 days. At the completion of the 120 h treatments, the entire surface of all panels (including the 3-h settlement panels in holding tanks) was inspected using a binocular microscope (10× magnification) and the number of settled larvae recorded.

Mechanisms of action: settlement disruption

To test our hypothesis that bubble streams disrupt settlement, we compared larval settlement success on blank ACR and FR-IS1100 panels (n = 6 per treatment combination, Fig. 1) placed in two tanks (with and without bubble streams) filled with Ciona (ca. 50 larvae L−1) and Crassostrea (ca. 10,500 larvae L−1) larvae. Each species was tested separately. Panels in the treatment tank were subjected to a medium intensity of bubbling (1.7 L h−1 cm−2) for a period of 24 h. During the Crassostrea trial, 20 L of Isochrysis galbana (approx. 10 million cells mL−1) was added to each of the tanks. After 24 h, panels were inspected under a dissecting microscope (×10 magnification) for the presence of settled larvae.

Field trials

Vertical surfaces

Twenty-four 300 × 400 mm panels were fixed vertically to four stainless steel frames suspended 1 m beneath a floating dock in Port Nelson (S 41°15′, E 173°16′). Twelve panels were constructed from concrete (CONC; a non-disclosed commercial marina pontoon formulation); the other twelve panels were constructed from acrylic professionally coated with a fouling release coating (FR-IS1100). Two of the experimental frames (each holding n = 3 CONC, n = 3 FR-IS1100) were continuously exposed to a bubble curtain treatment for 13 weeks. To achieve this, a 1.1 kw blower delivered 76 m3 air h−1 to two pairs (one pair per treated frame) of Hyotube fine bubble diffusers (see laboratory trials for blower and diffuser details). The diffusers were mounted 100 mm below the experimental panels (compared with 400 mm in the laboratory trials) to compensate for the strong tidal currents present in the field. Two treatment control frames, each holding three CONC and three FR-IS1100 panels, received no bubble treatment (Fig. 2).

Panels were visually inspected in situ after 1, 4, 6, and 9 weeks by a scientific diver familiar with biofouling assessments, and a categorical level of fouling (LoF; based on Floerl, Inglis & Hayden, 2005) was assigned. At the completion of the experiment (week 13), each panel was photographed (Canon PowerShot G16, 12.1 megapixels), the last visual LoF was assigned, and biofouling removed using a plastic scraper for biomass assessments.

Photoquadrat images of the experimental panels were analysed using the random dot method (Meese & Tomich, 1992) in Coral Point Count software (CPCe v4.1, Kohler & Gill, 2006), with 50 stratified random points overlaid on each image. Sessile taxa > 1 mm were identified to major taxonomic groups (barnacles, ascidians, bryozoans, tubeworms, oysters, mussels, hydroids, sponges, biofilm/bare space), and in the case of ascidians, further categorised based on their morphology (solitary vs colonial). A 2 cm margin around each experimental unit was excluded from analyses to avoid edge effects. Fouling biomass was measured following air drying to a constant weight (60 °C, 72 h).

Figure 2 Schematic of the experimental setup where concrete (CONC) and fouling release (Intersleek 1100SR, FR-IS1100) panels (300 × 400 mm) were treated with bubbles while fixed in a vertical orientation.

The same set-up was used for untreated panels (i.e., no bubbles), except frames were deployed without diffusers and the associated pipework.

Horizontal and angled surfaces

Between October 2018 and September 2019, the efficacy of continuous bubble streams was evaluated on horizontal and angled surfaces from a series of trials at the Devonport Naval Base, Auckland. A purpose-built raft was constructed using aluminium framing and polyethylene floats (1400 mm long, 700 mm diameter) for buoyancy (Fig. 3). The raft was fitted with 8 diffusers connected to a blower (K05 MS MOR 1.5kW; FPZ Blower Technology) via a series of hoses (40 mm internal diameter) and 2:1 hose connectors. Each diffuser hose line was fitted with a ball valve so that flow rates could be independently adjusted. For all trials, flow rates were set to correspond to the high flow used in the laboratory experiments (i.e., 2.6 L h−1 cm−2). The blower was also fitted with a variable frequency drive (Invertek Optidrive, Model ODE-2-12150-1KB1X) so that delivery of total air flow to the diffusers could be adjusted.

Figure 3 CAD drawing of the experimental raft (C) used in the field trials for horizontal (0°) and angled (22°) surfaces.

Experimental details for the three replicate trials (A) and the final treatment vs control (B) comparison (n=2) are shown. CONC, concrete; POLY, polyethylene; FR-IS1000, Intersleek 1000; FR-IS1100, Intersleek 1100SR.

A single diffuser was suspended 300 mm directly below each of 8 panels (500 mm × 750 mm) representing four surface types: concrete (CONC), black polyethylene (POLY), acrylic coated with International Paint FR coatings Intersleek 1100SR (FR-IS1100) and Intersleek 1000 (FR-IS1000). For each of the four surface types, one panel was fixed at 0° (flat) and 22° (angled). Due to space limitations, the first three trials had one replicate for each surface type:orientation combination (i.e., replication was achieved by undertaking three trials sequentially). Positions on the raft were randomly assigned for each trial. Following these trials, the accumulation of biofouling on POLY and FR-IS1000 panels with and without bubble treatment was assessed (Fig. 3). Panels were fixed in a flat orientation and were deployed for three months, with n = 2 panels for both treatments and controls.

After each deployment period the experimental raft was lifted from the water, panels were removed and photographed (Canon PowerShot SX280 HS, 12 megapixels), and new panels fitted. Photoquadrat images of the experimental panels were analysed using the random dot method in Coral Point Count software, with 200 stratified random points overlaid on each image. Sessile taxa >1 mm were identified to major taxonomic groups and their percentage cover estimated. A 2 cm margin around each experimental unit was excluded from analyses to avoid edge effects. Significant fouling development on the diffusers resulted in ‘shadows’ of treatment, and fouling could be observed directly above areas of the diffuser where this occurred. These patches of fouling were excluded from analyses (min = 0% of data points per panel, max = 47.0%, average = 9.7%). All panel images are provided in the Supplementary Material (S5).

Underwater noise

The use of a surface mounted blower and the release and subsequent collapse of microbubbles creates underwater noise. To gain insights into potential issues (e.g., impacts on marine mammals), underwater noise measurements were obtained from a bubble stream system deployed at Nelson Marina. The diffusers, powered by a 1.1 kw blower, were suspended 1 m below the water surface from a marina pontoon. Conditions within the marina during the trial were calm with an incoming tide of approximately 5 to 7 cm s−1. Sound traps (ST 300 STD units) were deployed on temporary moorings at 1-m and 3-m water depths adjacent to the diffusers, 3-m water depth 16 m from the diffusers, and at 3-m water depth 32 m from the diffusers. Acoustic recordings of the system were collected as follows: 10 min without the blower turned on (ambient); 7 min with the blower running, but not connected to the diffusers; 7 min with the blower running, and connected to the diffusers with maximum air flow; 7 min with the blower running, and connected to the diffusers with medium air flow (comparable to flow rates used during field trials); then 10 min without the blower turned on (ambient).

During each trial, a cabled hydrophone (Cetacean Research Technology CR1) was placed near to the diffusers to check for any extraneous noise contamination, such as an approaching vessel. Disrupted trials were discontinued and repeated after confirming the contaminating noise source was no longer detectable. Acoustic recordings were processed in MATLAB, where the power spectrum for each trial was calculated and plotted.

Data analyses

Laboratory trials

Generalized linear models (GLM, Dobson, 1990) were used to test how flow rate, settlement period and surface type affected the efficiency of the bubble treatment. For the scouring and disruption trials, we modeled individual counts of Ciona and Crassostrea using a GLM with negative binomial error distribution because the ratio between the residual deviance and the degrees of freedom showed that Poisson models were over-dispersed. For each trial, a separate model was constructed for each species.

Field trials

Vertical surfaces.

We performed an ordinal mixed effect model (Agresti, 2002) to test how surface type (CONC or FR-IS1100) and the bubble treatment affected the level of fouling (LoF) over the experimental period. An interaction term was included to test for a combined effect of the two experimental factors. Because repeated observations were made over a 13-week period, week number was added as a random effect in the models. A beta regression was used to test how surface type and bubble treatment affected the percent cover of macrofouling at the completion of the experimental period. A GLM (with Gamma error distribution) was used to explore factors affecting fouling biomass accumulation (assessed as dry weight). An interaction term was included in the last two models.

Horizontal and angled surfaces.

For the first three ‘rounds’ of the field trials, GLMs were used to test how surface type and orientation angle affected the efficiency of the bubble treatment in preventing biofouling accumulation on experimental panels (measured as biofouling percent cover). An interaction term was included in the models to test for a combined effect of the experimental factors. Because each round was undertaken during a different time period, experimental round was added as a fixed effect in the GLMs to account for seasonality. We separately modeled the percent cover of bare space, biofilm and macrofouling using a GLM with a binomial error distribution. Lastly, when the effect of bubble treatment was tested against control panels (two surface types, 0° orientation), the effects size was so large that the data were simply plotted and described.

All statistical analyses were performed within the ‘R’ statistical and programming environment (R Core Team, 2021).

Results

Laboratory trials

Mechanisms of action: scouring

Ciona settlement onto the FR-IS1100 panels proved unreliable as larvae ‘slid off’ the panel when tipped into the vertical position after the settling period, so these data were removed from the analyses. Settlement onto the acrylic panels was successful. Larval removal was significant for both the 3 and 120 h settlement periods when exposed to high bubble flow rates, with reduction of > 90% of settled larvae relative to the controls (Fig. 4, Table S3.1, p-value < 0.001). Medium and low flow rates were not different to the treatment controls, and we did not find an interaction of settlement period and flow rate.

Figure 4 Number of Ciona savignyi remaining on acrylic (ACR; A and B) and Crassostrea gigas (C and D) remaining on acrylic and fouling release (FR-IS1100) panels for both the 3 and 120 h settlement periods (testing the mechanism of scouring).

Air flow rates at the diffuser: Nil = no bubbles (treatment control), Low = 0.9 L h−1 cm−2, Med = 1.7 L h−1 cm−2, High = 2.6 L h−1 cm −2. N = 5 per treatment combination. Boxplots display the median, and the first and third quartiles (middle line and lower and upper hinges). The whiskers extend from the hinge to the largest or smallest value no further than 1.5 × the distance between the first and third quartiles. Ciona larval removal was significant for both the 3 and 120 h settlement periods when exposed to high bubble flow rates, with a reduction of > 90% of settled larvae relative to the controls (p-value < 0.001). Removal following medium and low flow rates was not significantly lower than the controls, and we did not find an interaction of settlement period and flow rate. For Crassostrea larval removal, we found a significant interaction among flow rate, surface type and settlement period (p-value < 0.001). All flow rates applied to the 120 h settlement panels were effective at removing larvae from the FR-IS1100 surface.

In contrast to the Ciona trials, there were no differences between the number of Crassostrea individuals remaining on the acrylic surface after treatment at any flow rate, regardless of settlement time. However, all bubble flow rates reduced the number of Crassostrea individuals (79–97% relative to the treatment controls) when individuals were settled for 120 h on the FR-IS1100 surface (Table S3.1, p-value < 0.05).

Mechanisms of action: settlement disruption

Panels exposed to continuous bubble streams resulted in nil settlement by Ciona over the 24 h treatment period, whereas settlement on control panels ranged from 16 to 115 and 64 to 851 larvae per panel for FR-IS1100 and ACR, respectively (Fig. 5). The continuous bubble stream also resulted in a significant reduction (p-value < 0.001) in Crassostrea settlement (average of 7.7 and 0.7 larvae per ACR and FR-IS1100 panel, respectively) compared with control panels (range = 163–774 larvae per panel). While statistically significant (Table S3.2), the treatment effect was not 100% effective for Crassostrea.

Figure 5 Number of (A) Ciona savignyi and (B) Crassostrea gigas larvae that settled onto blank fouling release (FR-IS1100) and acrylic (ACR) panels (200 × 150 mm) when exposed to a continuous bubble stream for 24 h (testing the treatment mechanism of settlement disruption). N= 6 per treatment combination.

Ciona and Crassostrea larval densities in the testing system were 50 and 10,500 L−1, respectively. Air flow rates at the diffuser: Nil = no bubbles (treatment control), Med = 1.7 L h−1 cm−2. Boxplots display the median, and the first and third quartiles (middle line and lower and upper hinges). The whiskers extend from the hinge to the largest or smallest value no further than 1.5 × the distance between the first and third quartiles. Panels exposed to continuous bubble streams resulted in nil settlement by Ciona over the 24 h treatment period. There was also a significant reduction (p-value < 0.001) in Crassostrea settlement compared with the controls. While statistically significant, the treatment effect was not 100% for Crassostrea.

Field trials

Vertical surfaces

Biofouling accumulation on control (non-bubbled) CONC panels occurred rapidly, with considerable fouling (LoF = 4) reached after 1 week, and extensive fouling (LoF = 5) evident after 1 month (Fig. 6C). During this same period, control FR-IS1100 panels were initially colonized by numerous small calcareous tubeworms (LoF = 3). After 3 months, other fouling taxa were also observed on the plates, including hydroids, tubeworms, bryozoans and ascidians. CONC panels receiving bubble treatment were first colonized by a slime layer with small numbers (1-3 per panel) of juvenile oysters. Over time, the number and size of oysters increased because once settled they were impervious to treatment, and other species (e.g., small mussels and hydroids) were found in low densities (LoF = 2). FR-IS1100 panels receiving the bubble treatment had a slime layer present after 4 weeks (LoF = 1) and remained devoid of macrofouling for the remainder of the trial. There was a significant interaction between surface type and treatment (all p-values < 0.001, Table S3.3), with higher LoF scores observed on the concrete control panels.

Fouling biomass reflected the LoF scoring (Fig. 6A), with highest levels recorded on the untreated CONC panels (15.6 g ± 1.6 g, mean ± 1SE), compared to 0.2 g ± 0.0 g on the treated FR-IS1100 panels. Oyster growth on the treated CONC panels contributed to an average biomass of 3.0 g ± 0.2 g, while the untreated FR-IS1100 panels averaged only 0.7 g ± 0.1 g per panel (SE = 0.1). Macrofouling percent cover closely matched the results for biomass (Fig. 6B). Biofouling dry weights and percent cover of macrofouling at the end of the experiment differed significantly between treatment and control panels, and a significant interaction between the surface type and treatment was observed (significant p-values < 0.001, Table S3.3).

Horizontal and angled surfaces

Continuous bubble streams applied to the four surface types (CONC, POLY, FR-IS1100 and FR-IS1000) fixed in two orientations (0° and 22°) kept macrofouling coverage low (average = 2%, SE = 0.01%), but biofilms developed on all surfaces (ranging between 27–100%; Fig. 7). Macrofouling taxa settling onto panels was restricted to barnacles and Pacific oysters. The temporal component was more important in determining the percent cover of bare space, biofilm and macrofouling than surface type or plate angle; round 3 was significantly different than rounds 1 and 2 for bare space, biofilm and macrofouling percent cover, and round 2 was significantly different than round 1 for bare space and biofilm (all p-values < 0.05, Table S3.4). Biofilm cover differed significantly between some surface types and angles, with highest cover observed on concrete surfaces fixed at 0° (Fig. 7).

When the performance of bubble streams was evaluated against controls, marked differences were observed (Fig. 8). Bubble treatment prevented the accumulation of macrofouling on FR-IS1000 and POLY panels, except on areas where the flow of bubble streams was disturbed by fouling on the diffusers of the experimental raft (excluded from the analyses). Bubble streams were also observed to reduce biofilm cover on FR-IS1000 panels. By contrast, the control panels for both surface types were colonized by a range of taxa, including barnacles, Pacific oysters, colonial and solitary ascidians, hydroids and filamentous algae (Fig. 9).

Figure 6 Effect of bubble streams applied to vertical panels.

Presented as: (A) dry weight (g) of biofouling (measured at the end of the experiment), (B) percent cover of macrofouling (at end of experiment), and (C) quantitative level of fouling (LoF) scores (observations made during the experiment). Total n = 6 per treatment combination, except treated concrete panels, where n = 3 due to compromised panels being excluded from analyses. Boxplots display the median dry weight and macrofouling percent cover on untreated (Control) and panels subject to continuous bubble stream treatment (Bubbled) after 13-weeks. The first and third quartiles of the data are shown (middle line and lower and upper hinges). The whiskers extend from the hinge to the largest or smallest value no further than 1.5 × the distance between the first and third quartiles. LoF scores were assigned by divers (weeks 1, 4, 6 and 9) and at the surface (week 13). LoF 0 = no visible fouling (including biofilm), 1 = slime fouling only, 2 = light fouling (1–5%), 3 = considerable fouling (6–15%), 4 = extensive fouling (16–40%), and 5 = very heavy fouling (41–100%). All models show a significant reduction of fouling on the bubbled treatment in comparison with the control (all p-values < 0.001), and a significant interaction between the bubbled treatment and control and the type of surface (p-value < 0.001).

Underwater noise

At 0.5 m distance from the diffusers, bubble curtain noise had a bandwidth of approximately 1190 Hz, spanning 10 Hz (minimum hydrophone sensitivity) to 1200 Hz (ranging between 81 and 111 dB re 1 µPa2 Hz−1) at maximum flow (Fig. S4). When flow was reduced to medium, the bandwidth dropped to approximately 835 Hz (spanning 15 Hz to 850 Hz), as did the spectral levels (to between 86 and 108 dB re 1 µPa2 Hz−1). The blower at the surface was generally non-detectable over the ambient underwater soundscape. At 16 m from the diffusers, underwater noise was partially attenuated (spectral levels peaked at approximately 86 dB re 1 µPa2 Hz−1), with a detectable frequency range between approximately 150 and 500 Hz. At 32 m from the diffusers, noise from the diffusers had attenuated out (i.e., no noise from the diffusers was detected over the ambient soundscape).

Figure 7 Percent cover of bare space, biofilm and macrofouling on experimental panels subjected to continuous bubble streams.

Four panel types (CONC = concrete, POLY = polyethylene, FR-IS1000 = Intersleek 1000, FR-IS1100 = Intersleek 1100SR) were oriented at two angles (A to D = 0°, plots E to H = 22°). The three trials/rounds (R1-3) were undertaken sequentially (n = 1 per experimental round). Bubble streams kept macrofouling coverage low, but biofilms typically developed on all surface types. The temporal component was more important in determining the percent cover of bare space, biofilm and macrofouling than the surface type or the position angle, evident by significant differences in percent cover of bare space, biofilm and macrofouling cover between rounds (p-value < 0.001). Biofilm cover differed significantly between some surface types and angles (p-value < 0.001), with highest cover observed on concrete surfaces fixed at 0°.

Figure 8 Percent cover of bare space, biofilm and macrofouling on experimental panels subjected to continuous bubble streams (Bubbled) alongside no treatment (Control).

(A) POLY = polyethylene, (B) FR-IS1000 = Intersleek 1000. N = 2 per treatment combination.Bubble stream resulted in reduced percent cover of macrofouling and biofilm on the FR-IS1000 surfaces and reduced macrofouling cover on POLY panels. Macrofouling was not detected on surfaces subjected to treatment, except on areas of the panels where bubble streams were interrupted by fouling on the diffusers or the experimental raft (excluded from analyses). By contrast, the control panels for both surface types were colonized by a range of taxa.

Figure 9 Representative images of biofouling on (A) Intersleek 1000 and (B) polyethylene panels with and without bubble stream treatment (duration = 119 days).

The hydroid fouling on the top left corner of the POLY treated panel was caused by fouling on the experimental raft interfering with bubble stream delivery. Similar ‘shadows’ were evident in the centre of panels immediately above sections of the diffusers that became fouled (see Supplementary Material S5).

Discussion

Building on the prior proof-of-concept research undertaken by Scardino, Fletcher & Lewis (2009), Bullard, Shumway & Davis (2010), Lowen et al. (2016), and Menesses et al. (2017), we have addressed key knowledge gaps around operational implementation of bubble streams/curtains to control biofouling accumulation on SAS. Our laboratory trials demonstrated that scouring of recently settled biofouling organisms was possible using bubble streams, including oyster larvae that had settled 5 days prior on a fouling release coating. We also demonstrated that larval settlement disruption was possible at flow rates that were insufficient to scour larvae from surfaces, supporting a new hypothesis for dual mechanisms of action of bubble streams. Field trials supported these findings, with panels remaining largely fouling-free after 2–3 months of deployment. These results are encouraging and indicate bubble streams are a potentially cost-effective, non-biocide-based long-term treatment for minimising biofouling on static submerged infrastructure.

Modes of action and factors affecting efficacy

The application of bubble streams over artificial surfaces limits biofouling development via at least two modes of action: scouring of settled larvae due to shear stress, and settlement disruption due to physical disturbance. For scouring, we found that surface type, flow rate and biofouling species present were all important determinants of treatment efficacy. Direct comparisons between FR-IS1100 and ACR panels were not possible for Ciona due to inconsistent (generally low) settlement on the fouling release panels. However, treatment efficacy was significantly higher on FR-IS1100 panels compared with ACR when bubble streams were applied to panels seeded with Crassostrea. Differences in efficacy due to bubble flow rates were more evident in the ACR panels for both species, indicating that thresholds for larvae removal (from shear stress) was higher for the ACR panels with higher surface tension. Of the two species, Crassostrea proved the most resistant to treatment.

Very few Crassostrea and no Ciona larvae recruited to virgin FR-IS1100 and ACR panels subjected to continuous bubble streams applied at the medium flow, while relatively high recruitment levels were observed on the control panels. Although not conclusive, these findings indicate that settlement disruption could be sufficient at bubble flow rates lower than that required to remove larvae that have settled for 3 h or greater. This supports the findings of Lowen et al. (2016), who found that flow rates capable of preventing 3-day old Ciona larvae from settling were insufficient to reliably remove 21-day old Ciona from petri dishes.

Underwater video footage of bubble streams interacting with the test panels revealed another potential mechanism that could be exploited to prevent fouling accumulation on surfaces fixed in a flat orientation. As bubbles came into contact flat panels, they would join to form large flat bubbles that were variable in size (ranging from 2–3 mm up to ca. 50 mm in diameter and several millimetres thick, Fig. 10). These trapped bubbles present a physical barrier to larval settlement. This phenomenon occurred under low, medium and high flow rates, and was evident on both the acrylic and fouling release panels. The formation and retention of large flat bubbles on the underside of flat structures could potentially be achieved through surface design and by delivering a mixture of small (providing settlement disruption and scouring mechanisms) and large bubbles (creating larger, persistent air pockets). For example, marina pontoons could be designed to trap air bubbles on the horizontal bottom surfaces, and essentially form a large air bubble to repel biofouling settlement.

Figure 10 The small (1 mm) bubbles from the diffusers were observed joining and forming much larger bubbles on acrylic and fouling release panels (200 × 150 mm) fixed in a horizontal orientation.

If this could be maintained, it could potentially be exploited to create a barrier between larvae and static coastal infrastructure.

For both the scouring and settlement disruption trials, the FR-IS1100 coating was found to be particularly amenable to treatment by bubble streams. For surfaces coated with fouling release paints, intermittent treatment (e.g., every 5 days) could be effective. If combined with the exposure to intermittent bubble streams, the use of fouling release coatings may be feasible for a broad range of vessel operational profiles (including vessels with extended lay-up periods), as biofouling removal would not solely be reliant on drag forces while vessels are underway. While application to concrete or polyethylene marina pontoons is technically feasible, consideration would need to be given to initial application costs, likelihood of damage from vessels and floating debris in these environments, and the longevity of coatings (see Hu et al., 2020).

Surfaces constructed from acrylic, polyethylene and concrete would likely require continuous treatment to maximise physical disruption of larval settlement, along with high flow rates to scour any settled larvae. Further, our observations of suboptimal performance against oyster larvae in the laboratory and field trials suggest that regions prone to calcareous fouling may require substantially higher bubble delivery rates than those applied in the present study to afford long-term protection. As observed in our field trials, once established, oysters can form treatment shadows as they change benthic boundary conditions on the surface, resulting in the establishment of taxa that would have otherwise been dislodged by the bubble streams. This has parallels with the encrusting bryozoan Watersipora subtorquata, which can colonise vessel hulls due to its tolerance to biocides and is then in turn colonised by less biocide-tolerant taxa (Floerl, Pool & Inglis, 2004). For bubble stream treatment approaches to be effective, near 100% efficacy against settlement is needed, otherwise periodic interventions (i.e., fouling removal by other means) would be required.

Due to the economic imperative to keep vessels free from marine growth (Schultz et al., 2011), studies have attempted to understand the forces required to dislodge biofouling while en route and during periodic maintenance (Crisp et al., 1985; Swain & Schultz, 1996; Callow & Callow, 2002; Finlay et al., 2002). This body of research has demonstrated that the adhesion strength of early microorganisms is weak and can be dislodged by a shear stress of around 1 Pa. Macrofouling larvae settle with an initial adhesion strength of around 0.1 MPa (Yule & Walker, 1984), and following metamorphosis, adhesion strength can increase greatly; e.g., around 1 MPa for barnacles (Crisp et al., 1985; Swain & Schultz, 1996). It is unsurprising then that vessels (or surfaces) that are cleaned/groomed more frequently have been found to require lower shear stresses to remove biofouling because the biofouling is likely to be less advanced/well adhered (Tribou & Swain, 2010; Tribou & Swain, 2015).

Menesses et al. (2017) examined the wall stresses required to keep panels free of fouling when exposed to a single bubble stream and found that levels greater than 0.01 Pa (10 mPa) were required, much less than that required to remove established fouling species. In the present study, air flow to the diffusers was manipulated to achieve a range of shear stresses acting on panels during the laboratory trials (176 ± 71 mPa to 415 ± 97 mPa for low and high flow rates, respectively, over panels held at 22°); these values are within the critical shear stress band required to remove larvae (Koehl, Crimaldi & Dombroski, 2013). It is therefore surprising that higher levels of treatment efficacy were not observed when bubble streams were applied to pre-settled Ciona and Crassostrea larvae in our laboratory trials.

In our study we were only able to estimate shear stress on plates held at an angle to the bubble stream, as perpendicularly orientated plates created a stagnation point at the plate surface where bubbles accumulated (Fig. 10) and made it impossible to calculate local velocity profiles using low resolution cameras. Therefore, it is possible that, due to the horizontal positioning of the panels and bubble interactions with panel coatings during trials, the shear stresses created at these flows were less than that required for dislodgement to occur. Alternatively, it is also possible that larvae were within the benthic boundary layer and therefore not subjected to the turbulent shear forces created at the boundary layer and free stream interface (Koehl & Hadfield, 2004; Massel, 1999). As bubble stream flow rates increase, the height of the benthic boundary layer would be expected to decrease (Cantwell, 1981; Crimaldi et al., 2002), possibly explaining why efficacy at the high flow rates was notably higher.

The influence of surface angle relative to the bubble stream delivery was not evident in our naval base field trials. In theory, shear stress should have been higher on the angled (22°) compared to the flat (0°) panels (Munson et al., 2013). It is likely that a lack of relationship between angle and increased treatment efficacy was masked by the high flows applied to the panels, i.e., shear forces generated over the flat panels were sufficient to remove (or disrupt) settling larvae.

Considerations for application in real-world settings

The uptake of novel approaches to manage biofouling accumulation on SAS will require low-maintenance, robust, fit-for-purpose and cost-effective systems that can be applied to a broad range of structure types. Given the significant cost associated with the production and installation of marinas and other marine infrastructure, there is a need to consider bubble delivery designs for not only new builds, but also for retrofitting to existing structures.

Bubble stream applications at greater water depths

Further studies are needed if bubble stream approaches are considered for managing biofouling on deep structures (e.g., oil rigs, wind turbines). With increasing water depths, air dissolution rates would increase (Woolf & Thorpe, 1991), and as bubbles rose to the surface, their volume would increase (doubling every atmosphere/10 m). Given that the relationships described in this manuscript relate to bubble sizes of around 1 mm diameter generated in < 1.5 m water depth, the efficacy of this approach at depth remains untested. Further work could include examining the fate and efficacy of bubbles released at depth, as well as overcoming some of the logistical challenges that could arise (e.g., bubble generation and system maintenance at depth).

Infrastructure, maintenance and running costs

The trials described in this paper used off-the-shelf components for generating and delivering bubble streams to experimental panels. Blower and diffusers costs for the continuous bubble treatment totalled around US$5000 (not including hardware for the raft) and treated approximately 3 m2. During the final trial period (ca. 4 months), an estimated 4300 kW h of electricity was used, equating to approx. US$230 of electricity usage per square metre (based on New Zealand power rates). For a 500-berth marina, with an estimated submerged surface area of submerged pontoons approximating 10,500 m2 (for simplicity, assuming 10 m finger lengths 1 m in width, 10 m distances between fingers, 2-m walkways down the middle, and all structures having a 0.5 m draft), this would result in over US$2.4 million in power costs per annum. If the experimental approach used in the present study was applied at an operational scale, electricity costs alone would be cost-prohibitive and existing alternatives (e.g., periodic fouling removal and capture by commercial divers) would be substantially cheaper. While more efficient systems could almost certainly be developed to run a much larger number of diffusers, and alternative sources of power could be considered (e.g., solar, wind, wave, and currents), capital and operational costs represent major challenges for uptake of this approach.

Keeping the diffuser free of fouling and sediment build-up was a challenge for both field experiments, and ongoing maintenance was required to ensure that the diffusers worked efficiently (Fig. 11). When left unchecked during the first field experiment on vertical panels, fouling growth on the experimental frames interfered with bubble delivery, resulting in fouling accumulation on the bottom third of the experimental panels. Similar treatment ‘shadows’ were also observed in the centre of many of the panels deployed horizontally. Any systems deployed in marine environments will also have to withstand periodic storms, tidal currents, collision with debris and vessels, and the corrosive nature of seawater. Ideally, treatment systems will be designed so that failure at single points does not result in complete failure.

Figure 11 Extensive biofouling growth developed on the experimental raft (A, dominated by barnacles) and diffusers (B, with extensive hydroid fouling) during each deployment.

Diffuser fouling resulted in sub-optimal delivery of bubbles to overlying panels, evidenced by discrete patches of macrofouling on otherwise clean (i.e., bare space or biofilm only) surfaces. Once formed, macrofouling patches persisted despite ongoing bubble stream treatment.

While challenges associated with diffuser fouling are significant, they could be addressed. Recent advances in antifouling technologies, such as copper cold spray (Vucko et al., 2012), could be applied to the diffusers, which will comprise a much smaller total surface area than the structures being treated. Systems could also be programmed to periodically dislodge sediment build-up by ‘blasting’ a higher velocity of bubbles through the delivery tubes, as is done in waste-water treatment oxidation ponds (Rosso, Larson & Stenstrom, 2008). Our laboratory trials also suggest that bubble delivery may not need to be continuous and could be timed so there is less disruption to marina users.

Underwater noise

Acoustic analyses revealed that the noise emissions were predominately low frequency (below 1200 Hz) and highly localised, propagating at levels above the ambient soundscape within 10s of metres rather than 100s of metres. Since the noise levels in this study we are the raw levels, the distance within which a marine mammal or fish may detect the noise over the ambient soundscape decreases further. However, it is important to note that this study involved the use of only two bubble diffusers (1 m length each). If the concept is to be applied at a large scale involving large numbers of diffusers (e.g., treating an entire marina), consideration would also need to be given to cumulative noise effects (see Pine, Jeffs & Radford, 2014).

The potential effect of treatment noise on humans also warrants consideration. For example, many marinas have rules controlling the noise from vessel engines, as well as devices such as radios and televisions. Such rules are typically in place to reduce the amount of disturbance to marina users and any nearby residents and businesses. Health effects arising from short and long-term exposure to noise are also possible. Noise created by bubbles collapsing at the surface is unlikely to pose a disturbance or health risk. However, noise created by pumps or blowers generating the bubble streams could be an issue, and mitigation measures (e.g., the use of sound-proofing materials, placement away from live-aboard vessels or residential areas) may be necessary.

Conclusions

Bubble streams have the potential to effectively limit biofouling development on SAS over extended periods of time. The development of cost-effective bubble delivery systems that perform under a large range of environmental conditions and that require low maintenance will be a formidable challenge. Future studies should aim to develop and test systems at an operational scale using approaches that can be retrofitted to existing infrastructure as most marine SAS have a service life of many decades. We expect an increased focus on the integration of biofouling management systems into the future designs of SAS if efficacy can be established at an operational scale. The development of bubble stream systems for installation below vessel berths would enable the use of non-biocidal fouling release coatings on slower vessels that currently need to predominantly rely on biocidal coatings.

Supplemental Information

Supplemental Information 1 Shear stress background and estimated values for the laboratory test system

Click here for additional data file.

Supplemental Information 2 Model organisms and spawning procedures

Click here for additional data file.

Supplemental Information 3 Outputs from statistical analyses

Click here for additional data file.

Supplemental Information 4 Underwater sound measurements

Click here for additional data file.

Supplemental Information 5 Images from field trials (horizontal surfaces)

Click here for additional data file.

Supplemental Information 6 Raw data from laboratory experiments

Click here for additional data file.

Supplemental Information 7 Raw data from field trials (horizontal surfaces)

Click here for additional data file.

Supplemental Information 8 Raw data from field trials (vertical surfaces)

Click here for additional data file.

Supplemental Information 9 Code for data analyses

Click here for additional data file.

The authors thank Sarah Strong (New Zealand Defence Force), the Dive team at the Devonport Naval Base, Don McKenzie (Northland Regional Council), Samantha Happy (Auckland Council), Kathy Walls (Biosecurity New Zealand), David Lamont (Bellingham Marine), David Hart, Brett Wallace and Aaron Lines (Akzo Nobel, International Paint), Dave Duncan (Port Nelson) and Bruce Lines (Diving Services New Zealand Limited) for logistical support and funding for aspects of this project.

Additional Information and Declarations

Competing Interests

Author Contributions

Data Availability

Matt K. Pine is an acoustic consultant who is employed by the Styles Group Underwater Acoustics. Matt volunteered his time to undertake the sound recordings reported in this project.

Grant A. Hopkins, Fletcher Gilbertson and Matt Pine conceived and designed the experiments, performed the experiments, analyzed the data, prepared figures and/or tables, authored or reviewed drafts of the paper, and approved the final draft.

Oli Floerl and Patrick Cahill conceived and designed the experiments, performed the experiments, authored or reviewed drafts of the paper, and approved the final draft.

Paula Casanovas analyzed the data, prepared figures and/or tables, authored or reviewed drafts of the paper, and approved the final draft.

The following information was supplied regarding data availability:

The factors and response variables for the laboratory trials (both Scouring and Disruption datasets), the field datasets (horizontal and vertical, respectively) and the code required to run the analyses are available in the Supplemental Files.

The code is also available in GitLab: https://gitlab.com/paula_casanovas/hopkins_etal_manuscript.

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
