# Peer review of "Continuous bubble streams for controlling marine biofouling on static artificial structures"

_PeerJ, doi:10.7717/peerj.11323_

## Round 0.1 · original submission · Major Revisions

The reviewers were generally complimentary of your work, but some revisions are necessary. The reviewer comments are straightforward and should be easy for you to follow. In particular, comments have been made to improve the clarity of the presentation. Of note are the lack of effective figure captions and statistical comparisons in the figures as well as the text. Some of the declarative statements about treatment comparisons cannot be made without proper statistical analyses supporting those statements. Please address these comments and the others from the 3 reviewers in your revision.

Reviewer 1 ·

Basic reporting

-- Introduction does a good job of providing background, context and highlighting key literature in the field.

-- Methods and Results
* Figure captions lack sufficient detail. A reader should be able to follow the manuscript narrative via figure captions, and not need to refer to the main text for key information. Figures 1–7 in particular are lacking in key background and contextual information. E.g., it needs to be made clear that Figures 4 and 5 relate to scouring and settlement disruption, respectively.

* Statistically significant differences should also be indicated in the figures.

-- Minor formatting and style corrections are needed throughout, e.g.,
* commas after e.g. and i.e.
* spaces between values and unit symbols.

-- Structure conforms to PeerJ standards.

-- Raw data are supplied.

Experimental design

-- The research question is well-defined, and the authors highlight key knowledge gaps the study attempts to address.

-- Methods
* Authors have been forthcoming in identifying limitations and logistical issues around experiments, and have explained the rationale behind work-arounds.

* Some content requires greater clarity or further information:
Lines 141–147: the order of operations is hard to follow, and could be explained more clearly.

Lines 182–184: some details are lacking on how taxa were assessed on experimental panels.

Validity of the findings

-- The data are robust and statistically sound.

-- Conclusions are well stated, and the authors do a good job of not over-interpreting the results.

Additional comments

The manuscript "Continuous bubble streams for controlling marine biofouling on static artificial structures" sets out to address some key questions around the use of aeration to prevent marine biofouling on submerged surfaces. Over the past ~ 10 years, aeration / bubble curtains has been touted as an environmentally-friendly (i.e., non-biocidal) tool that could be used to manage marine macrofouling on stationary structures and vessels, but this approach has received relatively little attention. The authors highlight the key papers in this field, and point out that there are some substantial knowledge gaps regarding the development and use of bubble curtains:
• Mechanism of action -- does turbulence disrupt organism settlement, or do shear forces scour larval recruits off?
• What air flow rates are effective at preventing macrofouling accumulation, and how is this influenced by organism and substrate type?
• How frequently should bubble streams be applied, and for how long?
• What's the potential for bubble stream systems to contribute to underwater noise pollution?

Through various experiments, the authors provide some insights towards these questions. The findings provide operational guidance to a first approximation, and will be of interest to municipalities, ports and marinas seeking to trial or test this biofouling management tool. This is also a solid contribution to the field of biofouling prevention technologies, and it will be of interest to researchers as it provides insights towards the mechanisms by which bubble curtains affect biofouling (e.g., via disruption and scouring).

The Introduction and the Discussion read well, the but figure captions lack sufficient detail, making it difficult for the reader to interpret and understand the Methods and Results.

Annotated reviews are not available for download in order to protect the identity of reviewers who chose to remain anonymous.

Reviewer 2 ·

Basic reporting

In a set of lab and field experiments, Hopkins et al. present an investigation of the effects of bubble streams on the scouring and settling disruption of marine organism settlement and growth on fouling panels. Panels composed different materials, compared with and without bubbling, and with and without anti-fouling coatings are tested. Sonic analyses were conducted in the field to determine the frequency bandwith and sound pressure emitted from the bubbling mechanism as a way of determining whether underwater sound might disrupt marine mammals and fishes. The authors put their results in the context of previous literature that investigates biofouling and air bubble/curtains effects, indicating that previous work has focused on scouring vs. settlement disruption. They also discuss the possibility and limitations of real-world bubbling systems scaled to the size of marinas.

Overall, this paper addresses the issue of how to prevent/minimize biofouling on static structures in the absence of shear forces that are produced when ships are propelled through the water, which are critical to modern anti-fouling coatings. With the global phasing out of tributyl tin, an element that is toxic to wide range of marine biofoulers, many of the new coating formulae rely on physical forces for efficacy. The investigators thus address an important limitation of modern coatings on stationary vessels and structures, focusing on the possible mechanistic effects of continuous bubbling on scouring and larval settlement disruption.

The manuscript seems to draw on appropriate literature to set the stage and context for their experiments and they discuss previous work that has examined the effects of bubbling on biofouling effectively.

The experimental approaches are fairly well described and appear sound. However, there are a number of aspects of the presentation of the data and analytical findings that need improvement.

Experimental design

The experimental and methodological designs were sound, but replication was limited in some instances (e.g., n=2 for treatments described in Fig 7). Simultaneous replication treatments for some experiments was limited by apparatus and the investigators repeated the experiment through time. There was mention of the importance of the temporal variable in relation to results, but the model results and statistics for all results were relegated to the Supplementary Materials. Below are some specific comments on issues that

Figures
The captions on some of the figures should to be revised to better describe what is being presented and highlight what a reader should take away from each. Figs 1-3 describe experimental apparatus and setup, and are reasonable. Although on Figure 3, it might be helpful if the labelling for FR-IS1000 and FR-IS1100 could be differentiated more clearly in another way (e.g., FR #1, FR #2). This was also problematic in Fig 6 and the associated Results section, where the single digit in the formal name was lost to this reader until the second or third pass. More obvious differentiation would make message more accessible and obvious. The authors might consider making this differentiation at lines 168-19 and/or 201-202.

Although opinions vary, my opinion is that figures that describe measurements/results should be able to stand alone from the main text and be fully interpretable. I found the data figures (Figs 4-7) to be lacking sufficient information. For example, Fig 4 (lab settlement results) present box plots but don’t provide any information about statistically relationships or differentiation. Text from the Results section (lines 270-275) indicate this differentiation, but then reference Table S3. Include and state statistical differentiation among box plot data distributions in Fig 4 – it would make for a far more compelling figure. Likewise, Figs 5-7 are summaries of the data, but could be improved if some notation (reflected in the figure captions or graph legends) specified how apparent differences/similarities. It is not clear to me that the information included in Fig 5 warrants a figure, a table (including statistical comparisons) may be preferable. Likewise, Figs 9 and 10 are only introduced in the Discussion section, so Supplementary Materials may be a more appropriate place for these.

Methods
The distance below the experimental fouling panels varied among lab and field experiments. In the lab the distance was 400mm, but 50mm in the vertical panel field experiments and 300mm in the horizontal and angled panel field experiment. There was no apparent rationale or justification for use of different distances from diffuser heads to the test panels, but it is logical to assume that concentration and vigor of bubbles is lessened with greater distance from diffuser. Some comment on why these depths were chosen, or justifying why they are comparable would be useful. Likewise, I had expected some discussion of the behavior or air bubbles underwater, specifically the effects of pressure and dissolution of gas into surrounding water has on bubble volume. Likely at the shallow depths considered in these experiments, the effects may be negligible, but if one considers scaling up bubbling systems to marina/harbor sizes, depth may become a factor.

Validity of the findings

The investigators draw conclusions that do not overstep their experimental findings, so thus seem valid. I provide some detailed suggestions in Experimental Design and General Comments to Authors that are meant to clarify and bolster their reporting. It strikes me that the manuscript could be more hard-hitting if some changes to the figure captions and main text were edited, as suggested.

Additional comments

Below are some detailed suggestions for edit that I think would strengthen the manuscripts and make it more accessible to readers.

Model organisms
Although the authors refer readers to details of Ciona and Crassostrea spawing and larval rearing in the Supplementary Materials, the question of larval competence is critical when considering settlement ability and success/failure. Some of this is addressed in Supplementary Materials, but a little more detail describing the importance of timing and competence on settlement success might be helpful in the main text.

Results
Although data analyses are described in Methods section on lines 242-262, including descriptions of multiple GLM models, there are no such results reported in the manuscript proper! The only numerical results that are reported in the paper are summary statistics: central tendencies and measures of variance. All modeling and statistical details are buried in Supplementary Materials, where many readers will never venture. The authors need to revise and expand their Results section to include more quantitative descriptions of their findings.

Vertical surfaces
Results are summarized in the text but it would be would be helpful and have more impact to reference a figure or consolidate the results in a table that includes the qualitative scoring as well as the % cover.

In the Methods and Results section, the authors may be able to better differentiate the field trials of horizontal and angled plates of various treatments if they described them as A) three trials (n=1 per treatment) replicated the experiment through time (i.e., the angled and flat panels summarized in Fig 6) and B) a separate experiment compared POLY with and without FR-IS1000 panels in the horizontal position (n=2 for each of four treatments). As is, it is confusing since a “final” trial is not similar to the previous experiment.

Figure 6 is the summary of a huge amount of work across a year, but the authors don’t seem to wring out as much information that the figure possesses. Again, some indications of statistical comparisons among treatment make the analysis more robust. Figure 7 reflects the results from two replicates, but there is no indication that the values reported are an average, since there is no measure of variance reported for the two replicates of each treatment. Experiment figures should all report sample sizes.

The apparently important temporal aspect of the horizontal/angled plate is mentioned, but glossed over and readers are referred to the Supplementary Materials for details. Given the generally well known importance of seasonality on marine invertebrate reproduction and settlement, any interaction between treatment and time should be further presented and discussed.

·

Basic reporting

The authors present a multi-stage study (several lab and several subsequent field trials) to fill knowledge gaps surrounding the efficacy and practical uptake of bubble streams to prevent the accumulation of biofouling on static artificial structures. They are successful at achieving these goals and at providing meaningful progress toward additional work that can build off of these results. There are, however, a few areas that need to be addressed to provide more clarity on the methods used and the overall readability of the manuscript.

Clarity:
- The Materials and Methods section appears to have been written out of sequence, with acronyms and abbreviations being used paragraphs or pages ahead of where they are formally introduced. Please review this entire section sequentially and adjust as appropriate. There are other issues with the description of the laboratory trials, but I will address those in the comments on Experimental Design.

- Lines 352-353: Sentence structure makes this statement unclear. Please clarify.

- Line 422: Should there be a "were" inserted between "forces" and "created?"

- Line 443: For clarity, please refer to the Naval Base experiment as "continuous bubble treatment." This makes it easier for a reader to understand to what you are referring, rather than making them refer back to the methods.

- Lines 458-459: For clarity, please refer to the Naval Base experiment as "continuous bubble treatment." This makes it easier for a reader to understand to what you are referring, rather than making them refer back to the methods.

Figures:
- The description for Figure 1 on line 111 states that two diffusers were placed on the bottom of the tank, but only one diffuser is visible in the figure. Presumably there is another diffuser in parallel with the displayed diffuser, but the figure should reference the paired diffusers or present a side view of the tank to display both diffusers. I think I understand the layout, but it is not clear and I should not have to guess when trying to understand the design.

- The caption for Figure 1 should describe the acronyms and abbreviations used in the figure, including FR-IS1100 and ACR (See Figure 4 for correct descriptions). Also in Figure 1, the diagram demonstrating the design for the scouring experiment does not show the five FR-IS1100 panels with Ciona. I understand from reading the rest of the manuscript that the Ciona failed to settle and remain on these panels and were, therefore, excluded from the analysis, but that fact comes much later than the introduction of the figure and description of the design. This leads to a lack of clarity when trying to review the figure while reading the description of the design. They do not add up and a reader doesn't find out why until much later. This can be addressed by including the panels in the figure and indicating that they were excluded from the analysis.

- The caption for Figure 2 should describe the acronyms and abbreviations used in the figure, including CONC and FR-1100SR (See Figure 4 for correct descriptions).

- The caption for Figure 3 should describe the acronyms and abbreviations used in the figure, including CONC, POLY, and FR-1100SR (See Figure 4 for correct descriptions).

- The caption for Figure 6 should describe the acronyms and abbreviations used in the figure, including CONC, POLY, and FR-1100SR (See Figure 4 for correct descriptions).

- The caption for Figure 7 should describe the acronyms and abbreviations used in the figure, including POLY and FR-1100SR (See Figure 4 for correct descriptions).

- The caption for Figure 8 should describe the acronyms and abbreviations used in the figure, including POLY and FR-1100SR (See Figure 4 for correct descriptions).

Experimental design

As indicated above, the materials and methods section appears to have been written out of sequence, making it difficult to follow what was being done. Please proofread this entire section to ensure that it flows sequentially.

Examples:
- In the description for the scouring experiment on lines 147-149, you describe the addition of an algal culture as a food source for the Crassostrea larvae, but do not mention anything about feeding of the Ciona larvae. In the next section describing the settlement disruption experiment, line 159 indicates that Ciona larvae are lecithotrophic and, therefore, did not require feeding. This should have come earlier.

- The abbreviation CONC is used on lines 170 and 174 without reference to what it means. It is not until the description of the field trials on line 200 that you define CONC as an abbreviation for concrete. This should have come earlier.

- The acronym for the foul release coating FR-1100SR was introduced on line 125 and is then used several times before being reintroduced on line 169. This is not necessary and suggests that these subsections were written separately and then combined without proofreading.

The subsection describing the scouring experiment is not clear and appears to contradict itself in several places. My understanding is that the panels were soaked with larvae for periods of 3 hours and 120 hours and then were subjected to 10-minute bubble treatments (or control) before being evaluated for retention. I am not clear on what is meant by the sentence on lines 143-145 and the references to 120-hour "treatment" on lines 147 and 150. I understood the 120 hours to be the settlement time before treatment and not the duration of the treatment, but the language on lines 147 and 150 is unclear and confusing. Please read this entire subsection and revise as necessary to ensure clarity.

Additional questions and comments about the methods:
- Please explain why the flow rates described on lines 118-120 were selected. Was this arbitrary or was this based on prior knowledge of previous work?

- Lines 135-136: Please provide the density of Ciona and Crassostrea larvae that were used in the settling tank.

- Lines 137-138: Please explain why the 3 and 120 hour time periods were chosen for settlement. Was this arbitrary or was this based on prior knowledge of previous work?

- Line 143: Please explain why 10 minutes was selected as the duration of the bubble treatment. Was this arbitrary or was this based on prior knowledge of previous work?

- Lines 149-150: Please provide more details on how the panels were inspected. Were the entire plates assessed for retained larvae?

- Lines 158-159: Please clarify if 20 L of Isochrysis galbana was added to both tanks (meaning it was split between the two tanks) or if 20 L was added to each of the tanks.

- Lines 176-177: Please describe how many divers were used and whether they independently assessed for Level of Fouling. How was the final category determined? By consensus?

Validity of the findings

The conclusions are well-stated. I would have liked to see a bit more here about the potential for future studies related to the formation and retention of large bubbles on the underside of flat structures and the influence of surface design. The idea is briefly mentioned on lines 371-374, but it would be useful to highlight here again with a bit more direction for future studies.

Additional comments

Overall, I think this work achieves the goals it set out to achieve and will contribute to the progress of this type of approach to preventing the accumulation of biofouling on these structures. The carryover benefits of minimally fouled infrastructure are important in reducing the spread of nonindigenous species as this can reduce the propagule supply and subsequent recruitment on vessels within these marinas.

I have a couple of additional comments that did not fit the other comment categories:

- Lines 442-450 contemplate scaling up the equipment and energy costs associated with treating an entire marina, but there is no reference to what that could look like at full scale. I suggest including the scaled up equipment and electricity costs for the entire marina that you worked in to give the reader a sense of how practical it would be and to contemplate what could be done to make those costs manageable.

- Are the authors aware of any local or regional sound restrictions within marinas? It would be helpful to put the underwater noise results into the context of what would be allowed to better interpret the results.

---

## Round 0.2 · Minor Revisions

Thanks for your efforts in revising your manuscript. There are some additional comments from the reviewers that I would like you to consider.

Reviewer 1 ·

Basic reporting

-- Introduction does a good job of providing background, context and highlighting key literature in the field.

Methods and Results
-- Figure captions have been revised and now provide sufficient detail that reflects the narrative of the manuscript.
-- Statistically significant differences are indicated accordingly.
-- Minor formatting and style issues have been corrected.
-- Structure conforms to PeerJ standards.
-- Raw data are supplied.

Experimental design

-- The research question is well-defined, and the authors highlight key knowledge gaps the study attempts to address.

Methods
-- The authors have been forthcoming in identifying limitations and logistical issues around experiments, and have explained the rationale behind work-arounds.
-- The revisions provide better clarity around the sequence of experimental steps and the characterisation of biofouling on experimental panels.

Validity of the findings

-- The data are robust and statistically sound.
-- Conclusions are well stated, and the authors do a good job of not over-interpreting the results.

Additional comments

The revisions have substantially improved the clarity of the manuscript.

This is a solid contribution to the field of biofouling prevention technologies, and it will be of interest to researchers as it provides insights towards the mechanisms by which bubble curtains affect biofouling (e.g., via disruption and scouring).

Annotated reviews are not available for download in order to protect the identity of reviewers who chose to remain anonymous.

·

Basic reporting

I am satisfied with the adjustments and additions that the authors made to address my comments. I have no further comments.

Experimental design

I am satisfied with the adjustments and additions that the authors made to address my comments. I have no further comments.

Validity of the findings

I am satisfied with the adjustments and additions that the authors made to address my comments. I have no further comments.

---

## Round 0.3 · accepted · Accept

Thank you for your willingness to continually improve your work based on reviewer comments.